# Behavioral and Physiological Response to Routine Thermal Disbudding in Dairy Calves Treated with Transdermal Flunixin Meglumine

**DOI:** 10.3390/ani12050533

**Published:** 2022-02-22

**Authors:** Tara Gaab, Mary Wright, Meghann Pierdon

**Affiliations:** New Bolton Center, Department of Clinical Studies, School of Veterinary Medicine, University of Pennsylvania, 382 West Street Road, Kennett Square, PA 19348, USA; tgaab@vet.upenn.edu (T.G.); marywri@vet.upenn.edu (M.W.)

**Keywords:** transdermal flunixin meglumine, calf, disbudding, behavior, pain

## Abstract

**Simple Summary:**

The practice of disbudding calves is common in the dairy industry, and the desire to mitigate pain caused by the procedure has resulted in questions as to whether all treatments are equally effective. The goal of this study was to investigate the effectiveness of a new product, transdermal flunixin meglumine, as part of a multimodal pain management protocol as compared to routinely used protocols. We determined that a pain management protocol utilizing transdermal flunixin meglumine and lidocaine was not significantly different than a protocol using meloxicam and lidocaine, or lidocaine alone, when comparing targeted calf behaviors and measuring salivary cortisol. This information can be used by veterinarians and producers to help guide them in choosing the appropriate pain management strategy for calves on their farms.

**Abstract:**

Transdermal flunixin meglumine was approved in 2018 to treat pain related to foot-rot in cattle, leading to the question of whether it would be effective as part of a comprehensive pain management strategy for disbudding. To investigate, calves were assigned to three treatment groups: 2% lidocaine cornual nerve block only (L), lidocaine nerve block +0.45 mg/lb (1 mg/kg) oral meloxicam (M), or lidocaine nerve block +1.5 mg/lb (3.3 mg/kg) transdermal flunixin meglumine (F) (*n* = 61). Ear flicking (*p* = 0.001), head shaking (*p* < 0.001), tail flicking (*p* < 0.001), interaction with the environment (*p* < 0.001), grooming (*p* < 0.01), posture changes (*p* < 0.05), and standing (*p* < 0.001) were impacted by the time relative to the procedure. Cortisol levels rose post procedure (*p* < 0.001). There was no difference in rates of behaviors or cortisol between treatments. These results indicate that calves showed alterations in behavior and cortisol in response to disbudding but not between treatments. We conclude that the pain management protocol for disbudding, which included transdermal flunixin meglumine with a lidocaine cornual nerve block, did not show significant differences from protocols using meloxicam with a lidocaine block, or a lidocaine block alone.

## 1. Introduction

The disbudding of calves is a common husbandry procedure in the dairy industry. Disbudding is deemed necessary for worker and animal safety, as the presence of large horns can result in difficulty moving and restraining animals and, additionally, may lead to injuries during normal social interactions between animals. Disbudding, the destruction of the corium or horn precursor cells, is widely preferred among veterinary professionals as opposed to dehorning at a more advanced stage of horn development; it is recommended to perform the procedure as early as possible. As a rule, 8 weeks old or less is recommended [1,2,3], with suggestions as young as 24 h [1]. Some countries require disbudding to be performed at 4 weeks old or less [4], thus, while most research on the topic of pain management for disbudding is focused on older calves (6 weeks up to 4 months), much of the present disbudding on farms is being performed on calves that are much younger. Therefore, we investigated the effects of three pain management protocols on calves that were less than 11 days old.

Pain mitigation is generally recommended in conjunction with disbudding [1,2,3], and over time the majority of stakeholders have indicated that they agree: pain relief should be provided at the time of procedure [5]. Most protocols include a local anesthetic with the recommended addition of a nonsteroidal anti-inflammatory drug (NSAID) [6]. Lidocaine (2%) is frequently used to block the cornual nerve that supplies sensation to the horn bud area, and it is preferable to procaine or topical anesthetics applied to the wound after the procedure, as those products have been reported to not be effective at relieving pain associated with dehorning [7,8]. Many studies have shown the benefits of adding meloxicam, either as an injectable or oral medication [9,10,11,12]. The advantages of using oral meloxicam are that it is relatively inexpensive and simple to administer to pre-weaned calves. The simplicity of a treatment is critical in achieving compliance by farm personnel, and this is one of the reasons transdermal flunixin meglumine has been an NSAID of interest ever since it was put on the market in the United States in 2018. Labeled to control pain associated with foot rot, and reduce fever associated with Bovine Respiratory Disease (BRD), the potential use of this product to control pain caused by elective procedures is intriguing due to its ease of use. Currently, its effectiveness for the provision of pain relief for disbudding is unknown.

Research investigating the effectiveness of transdermal flunixin meglumine for surgical castration in calves examined transdermal flunixin meglumine alone, but not as part of a pain management strategy that included a local lidocaine nerve block for pain mitigation. Kleinhenz et al. found that transdermal flunixin meglumine alone was not sufficient to decrease behavioral responses to castration, and it did not significantly affect other pain indicators including substance P, temperature measured at the medial canthus of the eye, and mechanical nociceptive threshold [13]. However, calves treated with transdermal flunixin at the time of castration had significantly lower levels of cortisol for 8 h post procedure compared to calves that received no NSAID treatment at castration. Kleinhenz et al. similarly investigated the effects of transdermal flunixin meglumine alone on disbudding, and they found that it did not influence substance P or infrared thermography levels [14]. There was a trend toward lower cortisol concentrations in flunixin-treated calves 90 min post-procedure, but it was not significant, and there was a significant increase in mechanical nociceptive threshold at the control site between the horn buds [14]. The suggestion that transdermal flunixin meglumine may reduce pain-related responses in calves, though not extensive, warrants further investigation in order to understand how it may fit into a pain management treatment plan that includes local anesthesia.

In this study, our goal was to determine if transdermal flunixin meglumine would provide pain relief as part of a multimodal pain management protocol in conjunction with a local anesthetic injection (F), as compared to a local anesthetic injection and oral meloxicam (M), or a local anesthetic injection alone (L), based on behavioral indicators and salivary cortisol concentrations. We hypothesize that there will be a significant treatment by time interaction, where calves provided an NSAID in addition to the local nerve block will exhibit differences in behaviors, and a decrease in salivary cortisol, post procedure compared to the calves given lidocaine alone.

## 2. Materials and Methods

The study was conducted from May to July 2019 at two Holstein dairies in Cochranville, Pennsylvania. Farms were enrolled if they were keeping their heifers onsite for at least 14 days after birth while also housing calves individually. All calves were enrolled at less than 11 days of age, with the average age 5 ± 2.3 days.

### 2.1. Treatments

A total of 61 calves were randomly assigned using a random number generator (Microsoft Excel, Microsoft Corporation, Redmond, WA, USA, 2018) to one of three treatment groups: 10 mL 2% lidocaine cornual nerve block only (L, *n* = 24), lidocaine nerve block +0.45 mg/lb (1 mg/kg) oral meloxicam (M, *n* = 20), or lidocaine nerve block +1.5 mg/lb (3.3 mg/kg) transdermal flunixin meglumine (F, *n* = 17) (Banamine Transdermal, Merck Animal Health; Madison, NJ, USA). Sample size differed for each treatment due to the order of randomization and the number of calves available during the study period. Sample size was calculated based on the ability to detect a predicted effect size for the salivary cortisol of 1 µg/dL with α = 0.10 and power = 0.85, which resulted in a sample size of at least 16 calves in each group.

### 2.2. Disbudding Procedure

Up to five calves were treated, disbudded, and observed per session. The calf’s head was restrained using a rope halter, and 5.0 mL of 2% lidocaine was administered to each side of the head for a cornual nerve block in all calves. The calves in Group M were administered 0.45 mg/lb (1 mg/kg) of Meloxicam orally by hand, and the calves in the other two treatment groups received a sham Meloxicam treatment by way of the experimenter allowing the calves to suck on their hand for several seconds. Calves in group F received 1.5 mg/lb (3.3 mg/kg) of transdermal flunixin meglumine (average = 3.0 mL) down the topline at the level of the withers. The location along the topline was chosen to prevent the flunixin meglumine from being licked off by the calf, and per the label, was not applied if the conditions were wet. All calves in other groups received a sham treatment of saline and red food coloring to mimic the appearance of the transdermal flunixin meglumine and keep the behavioral observer blinded. All treatments for each calf were administered consecutively and in the same order. The experimenter waited 10 min after the time of lidocaine injection before beginning the disbudding procedure. All calves were disbudded using a Portasol^®^ (Portasol^®^ USA, Elmira, OR, USA) butane dehorner. After disbudding, each calf was treated with a topical aluminum aerosol bandage and a permethrin-based wound spray.

### 2.3. Saliva Collection and Cortisol Analysis

Four total saliva samples per calf were collected: pre-treatment, 30 min post-procedure, 2 h post-procedure, and 24 h post-procedure. An approximately 2.5 cm by 2.5 cm piece of cotton was used to swab the calf’s mouth until fully moistened. The swab was placed in a 10 mL syringe and immediately refrigerated. The samples were then transported to a centrifuge, placed in a 50 mL conical tube, and spun for 2 min at 1100 rpm. This allowed the saliva to collect at the bottom of the tube. The samples were then stored at −80 °C until testing. Salivary cortisol concentration was measured via ELISA, using the High Sensitivity Salivary Cortisol Enzyme Immunoassay Kit (Salimetrics, State College, PA, USA).

### 2.4. Behavioral Examinations and Behavioral Analysis

Each calf was observed for 5 min by a single, blinded, observer prior to being handled for the initial saliva collection. Each calf was then observed for 10 min immediately post-procedure. Starting 60 min after the procedure, each calf was observed for 5 min at a time, every 25 min, over the span of 4 h. The following day, 24 h post-procedure, each calf was again observed for 5 min at a time, every 25 min, over a span of 2 h. The ethogram shown in Table 1 was used for the behavioral observations. Point behaviors were recorded as all occurrences of each, whereas standing was recorded as having occurred in the observation window or not occurred. The observer remained blinded to treatment throughout the study.

### 2.5. Statistical Analysis

All statistical analysis was performed using STATA v.15 (StataCorp. 2017. Stata Statistical Software: Release 15. College Station, TX, USA: StataCorp LLC, TX, USA). Point behaviors were consolidated into 4 observation periods, one prior to the procedure (1 interval, 5 min), one immediately after the procedure (1 interval of 10 min), one starting at 60 min after the procedure and continuing for 4 h (8 intervals of 5 min each, 40 min total), and one for 2 h and 24 h after the procedure (5 intervals of 5 min each, 25 min total). The behaviors were summed for each period and then divided by the number of minutes of observation to create a rate (behavior per minute) for that behavior for each period. As standing was recorded as a binary, present or absent during the observation interval, the number of times the calf was found standing was summed and then divided by the number of observation intervals.

All behavior rates and cortisol values were analyzed using mixed effect regression models, with the rate of each behavior, or cortisol concentration, as the outcome, and treatment, time, the treatment by time interaction, and farm as fixed effects, and the calf as random effects. Post hoc tests for comparisons between multiple levels were performed, and a Bonferroni correction was applied to account for multiple comparisons. Visual inspection of plots of residuals did not show any obvious deviations from homoscedasticity. Results are reported as least square means and standard error unless otherwise noted. *p* < 0.05 was treated as significant.

## 3. Results

### 3.1. Behaviors

There were no significant treatment by observation period interactions (*p* > 0.05) for any of the behavior rates (*p* > 0.05) or standing (*p* > 0.05) (Table 2).

#### 3.1.1. Ear Flicking

There was a significant impact of observation period on ear flicking per minute (*p* = 0.001), with significantly less ear flicking directly after the procedure (0.07 ± 0.02) compared to 5 min before (0.18 ± 0.04) (*p* < 0.05), and 24 to 26 h after (0.28 ± 0.06) (*p* < 0.01). During the period 24 to 26 h after the procedure, calves also exhibited more ear flicking per minute compared to the 10 min directly after the procedure (0.07 ± 0.02) (*p* < 0.001), and hours 1 to 5 after the procedure (0.10 ± 0.01) (*p* < 0.001) (Figure 1).

#### 3.1.2. Head Shaking

There was an impact of observation period on head shaking (*p* < 0.001), with significantly more head shaking per minute right after the procedure (0.47 ± 0.07) compared to 5 min before the procedure (0.21 ± 0.03) (*p* < 0.01) as well as 1 to 5 h after the procedure (0.14 ± 0.02) (*p* < 0.001), and 24 to 26 h after the procedure (0.25 ± 0.03) (*p* < 0.01) (Figure 1).

#### 3.1.3. Tail Flicking

There was a significant impact of observation period on tail flicking (*p* < 0.001), with more tail flicking in the 5 min before the procedure (1.16 ± 0.17) compared to post procedure (0.65 ± 0.09) (*p* < 0.05), 1 to 5 h after the procedure (0.23 ± 0.04) (*p* < 0.001), and 24 to 26 h after the procedure (0.62 ± 0.07) (*p* < 0.01). Calves performed less tail flicking 1 to 5 h after the procedure compared to directly after disbudding (*p* < 0.001) and 24 h later (*p* < 0.001) (Figure 1).

#### 3.1.4. Interaction with the Environment

We found a significant influence of period of observation on interaction with the environment (*p* < 0.001), where calves interacted less per minute with their environment in the period 1 to 5 h after the procedure (0.16 ± 0.02) compared to the period before disbudding (0.45 ± 0.06) (*p* < 0.001), the period directly after disbudding (0.30 ± 0.05) (*p* < 0.05), and 24 h later (0.29 ± 0.03) (*p* < 0.001) (Figure 1).

#### 3.1.5. Grooming

There was a significant effect of observation period on the rate of grooming (*p* < 0.01). We found a decrease in grooming in the period 1 to 5 h post procedure (0.20 ± 0.03) compared to immediately post procedure (0.34 ± 0.05) (*p* < 0.01) and 24 h later (0.28 ± 0.03) (*p* < 0.05). (Figure 1).

#### 3.1.6. Head Rubbing

We found no effect of observation period on the rate of head rubbing (0.06 ± 0.01) (*p* = 0.44).

#### 3.1.7. Standing and Posture Changes

Posture changes were significantly influenced by observation period (*p* < 0.01). The rate of posture changes was lower 1 to 5 h post procedure (0.04 ± 0.01) compared to immediately post procedure (0.06 ± 0.01) (*p* < 0.001) and 24 h later (0.05 ± 0.01) (*p* < 0.05) (Figure 1). There was a significant impact of observation period on the proportion of observations where the calf was standing (*p* < 0.001), with calves observed standing more post procedure (0.99 ± 0.02) compared to before the procedure (0.56 ± 0.06) (*p* < 0.001), the period 1 to 5 h after the procedure (0.19 ± 0.02) (*p* < 0.001), and 24 to 26 h later (0.35 ± 0.02) (*p* < 0.05). Calves were also found standing at a lower proportion of observations 1 to 5 h after disbudding compared to the 5 min before the procedure (*p* < 0.001) and 24 to 26 h after disbudding (*p* < 0.001) (Figure 2).

### 3.2. Salivary Cortisol

There was no significant treatment by time of sampling interaction on the salivary cortisol concentration (*p* > 0.05), and there was no significant impact of treatment: Lidocaine 1.61 µg/dL ± 0.07, flunixin meglumine 1.71 µg/dL ± 0.07, and meloxicam 1.75 µg/dL ± 0.09 (*p* = 0.34) (Table 3).

There was a significant impact of the time post disbudding on the salivary cortisol concentration (*p* < 0.001), with calves having higher salivary cortisol 2 h post procedure (1.9 µg/dL ± 0.06) compared to prior to disbudding (1.6 µg/dL ± 0.06) (*p* < 0.001), and higher than 30 min post procedure (1.5 µg/dL ± 0.04). Salivary cortisol at 24 h (1.8 µg/dL ± 0.06) was higher than pre-procedure (*p* < 0.05), and higher than 30 min after disbudding (*p* < 0.001) (Figure 3).

## 4. Discussion

The behaviors observed and recorded in this study were chosen based on studies that document significant alterations in the behaviors after disbudding, indicating that they are possibly associated with pain [11,15]. The age of the calves examined in this study aligns closely with the age that calves are disbudded in the field, although it is younger than the calves in most disbudding and dehorning experimental studies [9,10,11,14,15,16,17,18,19]. The behavioral repertoire of calves less than two weeks of age may differ from older calves, and, therefore, direct comparisons of behavioral analyses should take this into account. However, we did find an impact of the disbudding procedure on multiple behaviors. Therefore, while calf pain-related behavior (at less than two weeks old) is not as well-described in the existing disbudding literature, our findings suggest that these calves are demonstrating significant behavioral responses to disbudding.

Our study revealed a significantly higher rate of head shaking post-procedure, with the behavior returning to pre-procedure levels for the next four hours and continuing at a low rate 24 h later. These findings support the evidence observed in the literature, which indicates that there is significantly more head shaking following disbudding and dehorning [8,11,18]. In Heinrich et al. [11], an increase in head shaking was evident after the procedure in both calves that received lidocaine only and calves that received lidocaine along with injectable meloxicam. Graf & Senn [18] also found that head shaking increased after disbudding, regardless of whether calves received a local anesthetic treatment, compared to the low levels of head shaking observed after a sham dehorning procedure. Graff & Senn [18] and Heinrich et al. [11], along with our research, suggest that head shaking is a behavior that is influenced by the disbudding procedure itself, and will be expressed by calves that have received local anesthetic and NSAID treatments. In contrast to studies that found a significant reduction in head shaking in NSAID-treated calves compared to those that received only lidocaine [11,18], Milligan et al., using calves that were less than 14 days of age, did not report any difference between lidocaine and ketoprofen-treated calves and those that received only lidocaine, and this is similar to our findings [20]. The difference in age between studies may explain why head shaking is a behavior that is always observed after disbudding but does not always differ significantly between treatments.

In contrast to head shaking, we noted a decreased rate of ear flicking immediately post procedure, and an increased rate 24 to 26 h later. This is different from some studies that determined ear flicking to be a significant disbudding-related behavior, reporting a significant amount of ear flicking post-procedure, more so in calves that did not receive NSAID treatments in addition to local anesthetic treatments [11,15]. However, Milligan et al. [20], who examined calves younger than two weeks old (as we did), similarly found no appreciable difference in frequency of ear flicking when comparing calves that received lidocaine and ketoprofen versus those that only received lidocaine. As part of our experimental design, we applied fly spray to the affected area at the end of the disbudding procedure in order to provide the standard of care by preventing infection and fly strike. This is another potential cause of the pattern of ear flicking that we recorded, because the spray was not reapplied 24 h after the disbudding procedure, and this is when ear flicking resumed.

Similar to Herskin et al., no difference in the frequency of tail flicking between treatment groups was observed [21]. However, in contrast to Herskin et al., we did observe a significant decrease in the frequency of tail flicking in the period 1 to 5 h after the disbudding procedure, while Herskin found an increase in the total number of tail flicks after disbudding [21]. We suspect that tail flicking may be related to other environmental factors, and, as a consequence of this, it is not a reliable behavioral indicator of pain associated with disbudding, although it may be reliable with castration or tail docking.

Calves demonstrated a decreased rate of environmental interactions after the disbudding process compared to pre-procedure rates. There was no difference between treatment groups, but the decrease in rate of environmental interactions may indicate the calves’ lack of motivation to explore their environment after a painful procedure. Similarly, the amount of grooming 1 to 5 h post-procedure was significantly lower than immediately after and 24 to 26 h after the procedure. This decreased behavior is reflective of the overall decreased activity calves exhibited 1 to 5 h after disbudding, and it may be further evidence of their response to the painful procedure.

Grooming was a behavior that was also of interest due to the transdermal nature of the flunixin meglumine product, as calves that have been observed self-licking were reported to have decreased absorption of the active ingredient, although no decrease in effectiveness was reported [22]. We also wanted to account for any potential behavioral reaction the product might cause, because mild application site reactions have been reported [22]. Despite those reports, we found no treatment effect or treatment by time interaction. The calves in this study did not show significantly different rates of grooming in the period immediately after application of the product, nor did they exhibit different rates of grooming between sham topical application or the actual product.

The rate of head rubbing behavior was not significantly different between treatment groups, and it did not change significantly after the disbudding procedure. The relationship of head rubbing to pain associated with dehorning is not yet clear in the literature, and while some studies have reported it to be a significant finding associated with disbudding [16], others have found, as we did, that this behavior is displayed at a very low frequency, and it does not appear to change after disbudding [15]. Differences in housing, environment, calf age, and method of disbudding may all be factors in whether head rubbing is a commonly expressed behavior.

The significantly greater proportion of standing observations that were observed in the immediate post-procedure period most likely reflect the disbudding procedure’s requirement that the calf be standing and restrained. There is no difference in standing behavior or in postural changes between treatment groups; both behaviors are at significantly lower proportions 1 to 5 h after the immediate post-procedural period. The combination of significantly lower standing observations and postural changes may indicate a degree of pain relief [11,23], in this case provided by the lidocaine. However, activity behaviors exhibited by young calves should be interpreted with caution, as some studies have suggested that decreased lying behavior and increased activity is a sign of well-managed pain [24], while others suggest that increased lying and decreased activity are indicative of comfortable resting behavior [11]. When comparing the post procedure behaviors to the higher proportion of standing and higher rate of postural changes before the disbudding procedure, it is more apt to interpret the significant decrease in activity, in this study, to be a response to a painful and stressful event.

To avoid additional stress in addition to the restraint and disbudding procedures, salivary cortisol was used as a physiologic indicator for stress, as has been performed in other neonatal calf studies [25], rather than measuring plasma cortisol. Due to the time lag that has been reported between peak plasma cortisol and peak salivary cortisol [26], in combination with the known absorption and half-life of transdermal flunixin meglumine causing a peak in serological concentrations of flunixin at 2.14 h [27], collection time points at 30 min, 2 h, and 24 h were hypothesized to be the most likely times to observe significant differences in salivary cortisol concentrations after disbudding, based on previous transdermal flunixin meglumine disbudding studies [14]. There was no significant difference in salivary cortisol levels between treatment groups. However, at two hours post disbudding there was a significant increase in cortisol among all calves, indicating that the stresses associated with handling and the procedure were enough to raise levels significantly in the saliva.

To our knowledge, this is the only study to date that has examined transdermal flunixin meglumine as part of a pain management protocol including local anesthetic relief. While the data indicated that the procedure itself had an impact on the behaviors that we assessed, the different treatments did not, which could suggest that the effects of the procedure on behavior may have masked any subtle differences between treatments. It is also possible that the lidocaine, which was common to all three treatments, was the overriding pain mitigation technique impacting the behavior changes that may have been present in the first five hours after the procedure, as cornual nerve blocks have been reported to have a duration of up to five hours or longer [28]. Therefore, based on our results, we conclude that transdermal flunixin meglumine with a local lidocaine cornual nerve block did not show significant differences in comparison to protocols which manage pain associated with disbudding by using meloxicam with a local lidocaine block, or a local lidocaine block alone.

Looking forward, future research evaluating the effectiveness of transdermal flunixin meglumine for pain relief as part of a multi-modal protocol should include administering the product at other time points, as, at these points, the effects of lidocaine would be expected to wane and pain mitigation would be needed. Additionally, other pain-related physiologic responses, such as mechanical nociceptive threshold and ocular temperature, would be valuable additions in order to understand the effectiveness of the product. Finally, investigating how disbudded calves, treated with transdermal flunixin meglumine, perform in behavioral tests, such as cognitive bias tasks, may uncover subtle differences between treatments.

## 5. Conclusions

Our study found significant behavioral changes in calves (less than 11 days of age) after the disbudding procedure. Specifically, calves performed more head shaking while decreasing ear flicking, tail flicking, grooming, posture changes, standing, and interactions with the environment after the procedure. Additionally, we found a significant increase in salivary cortisol levels which started two hours after disbudding. While no difference in any behaviors, or in salivary cortisol, was observed between treatment groups, the behavioral impacts of the procedure show that the procedure is painful, and that behavioral differences can be appreciated in calves that are less than two weeks old. Future research, utilizing additional pain assessment strategies, may shed light on any differences between treatments. We conclude that no differences were observed in behavior, or cortisol response, when we examined disbudded calves given transdermal flunixin meglumine with a local lidocaine block, compared to calves given oral meloxicam with a lidocaine block, or calves given a lidocaine block alone.

## Figures and Tables

**Figure 1 animals-12-00533-f001:**
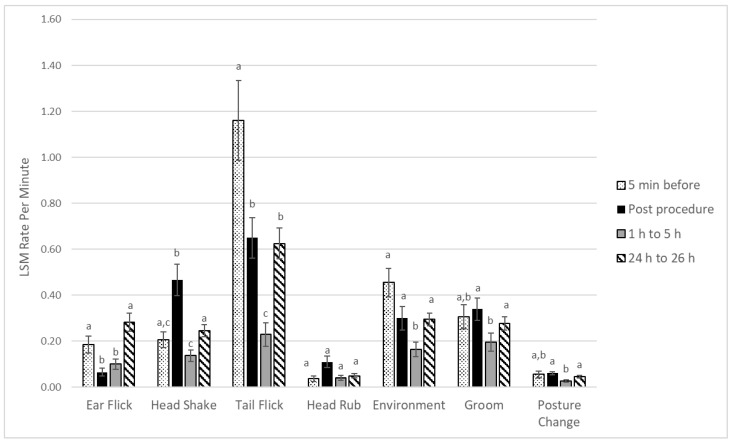
Effect of observation period. Results are presented as least square means and standard error based on mixed effect linear regression models, with each behavior as the outcome. Treatment, time, and the treatment by time interaction were included as fixed effects with the calf as random effects. The different superscripts indicate a statistically significant difference (*p* < 0.05) between periods for the rate of that behavior.

**Figure 2 animals-12-00533-f002:**
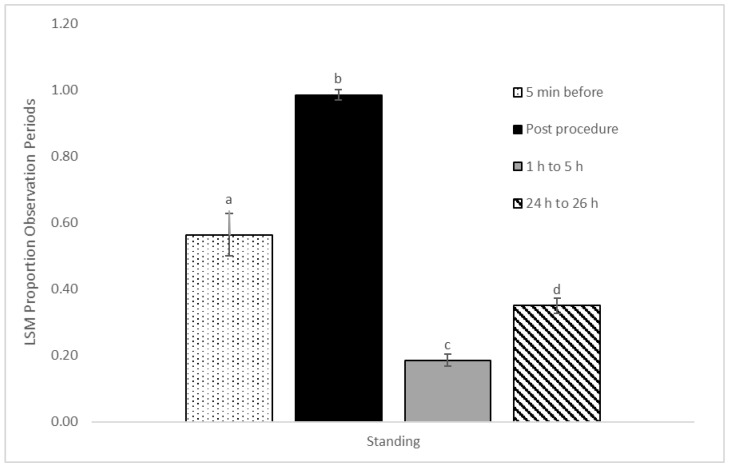
Effect of observation period on proportion of observations standing. Results are presented as least square means and standard error based on a mixed effect logistic regression model, with treatment, observation period, the treatment by period interaction, and farm as fixed effects, and calf as the random effects. As standing was recorded as present or absent during the observation interval, the number of intervals where the calf was found standing was summed and then divided by the number of observation intervals for that period. Different superscripts indicate a statistically significant difference of *p* < 0.05.

**Figure 3 animals-12-00533-f003:**
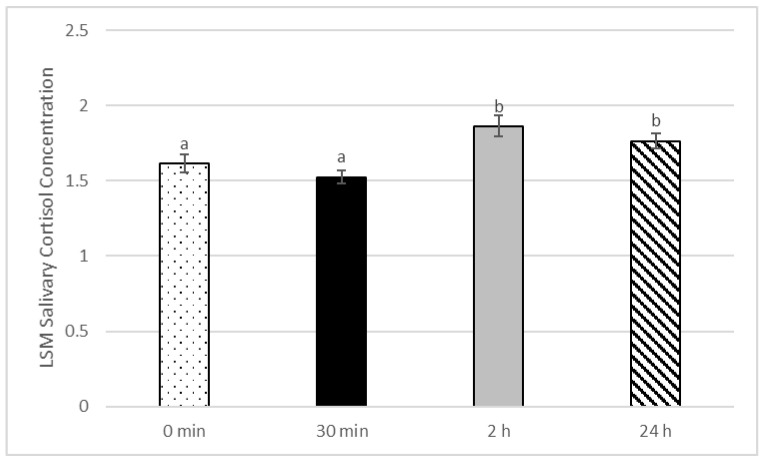
Impact of sampling time on salivary cortisol concentration. Results are the least square mean and standard error based on a mixed effect logistic regression model, with treatment, sampling time post disbudding, the treatment by time interaction, and farm as fixed effects, and calf as the random effects. Different superscripts represent a statistically significant difference between sampling time points of *p* < 0.05.

**Table 1 animals-12-00533-t001:** Ethogram of calf behaviors and their respective definitions.

	Behavior	Description
State	stand	All four feet on ground, with no other body part touching the floor
Point Behaviors	ear flick	Movement of ears forward and backward rapidly, without head movement. Each time the ears move, it is defined as a single ear flick. ^1^
	head shake	Movements of the head from side to side quickly. Recorded as a single behavior until head movement stops. ^1^
	tail flick	Movement of tail back and forth rapidly. Recorded as a single behavior until tail movement stops. ^1^
	head rub	Lifting of the hind leg to scratch the head with foot or rubbing of head against hutch. ^1^
	grooming	Turning head and neck to lick self on side or back. Recorded as a single behavior until licking stops.
	environment	Sham nursing on hutch sides or bars. Recorded as a single behavior until nursing stops.

^1^ Heinrich et al. [11].

**Table 2 animals-12-00533-t002:** Effect of treatment. Results are presented as least square means and standard error based on mixed effect linear regression analysis. The behaviors were measured for 4 periods, one prior to disbudding (1 interval, 5 min), one immediately after disbudding (1 interval of 10 min), one starting at 60 min after disbudding and continuing for 4 h (8 intervals of 5 min each, 40 min total) and one for 2 h and 24 h after disbudding (5 intervals of 5 min each, 25 min total). The behaviors were summed for each period and then divided by the number of minutes of observation to create a rate for that behavior. As standing was recorded as present or absent during the observation interval, the number of intervals where the calf was found standing was summed and then divided by the number of observation intervals.

Behavior	Lidocaine	Flunixin	Meloxicam	*p*-Value
stand	0.51 ± 0.03	0.53 ± 0.03	0.52 ± 0.03	0.77
posture change	0.04 ± 0.01	0.05 ± 0.01	0.05 ± 0.01	0.55
ear flick	0.19 ± 0.03	0.13 ± 0.02	0.16 ± 0.03	0.64
head shake	0.25 ± 0.03	0.21 ± 0.03	0.33 ± 0.06	0.77
tail flick	0.68 ± 0.10	0.53 ± 0.08	0.79 ± 0.15	0.52
head rub	0.04 ± 0.01	0.07 ± 0.02	0.07 ± 0.01	0.44
environment	0.33 ± 0.05	0.25 ± 0.04	0.33 ± 0.04	0.10
groom	0.32 ± 0.06	0.23 ± 0.03	0.29 ± 0.04	0.21

**Table 3 animals-12-00533-t003:** Salivary cortisol concentration by treatment and time. Results are presented as least square means and standard errors based on a mixed effect regression model, with cortisol concentration as the outcome, treatment, time, treatment by time interaction, and the farm as fixed effects, and the calf as random effects. Cortisol concentration was measured at 4 time points, one prior to disbudding (Time = 0), again 30 min after disbudding, at 2 h, and finally at 24 h post procedure. There was no significant effect of treatment, and no significant treatment by time interaction.

Time Post Procedure	Lidocaine (µg/dL)	Flunixin (µg/dL)	Meloxicam (µg/dL)
0	1.51 ± 0.05	1.67 ± 0.10	1.67 ± 0.12
30 min	1.50 ± 0.07	1.50 ± 0.06	1.58 ± 0.08
2 h	1.78 ± 0.12	1.91 ± 0.11	1.90 ± 0.11
24 h	1.67 ± 0.06	1.77 ± 0.09	1.85 ± 0.10

## Data Availability

The data presented in this study are available on request from the corresponding author.

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
