# Peer review of "Behavioral and Physiological Response to Routine Thermal Disbudding in Dairy Calves Treated with Transdermal Flunixin Meglumine"

_animals, 2022, doi:10.3390/ani12050533_

Round 1

Reviewer 1 Report

Well designed study and well reported. Minor amendments suggested here:

ABSTRACT

Should contain timepoints for measures taken.

INTRO

Line 62: Change ‘pain strategy’ to ‘pain management strategy’.

Line 65: Infrared thermography isn’t a pain indicator but a method for measuring temperature. Suggest replacing with the outcome that infrared thermography was used to measure.

METHODS

Lines 96-97: How was sample size determined and why are animal numbers unequal between treatment groups? Were treatment groups balanced for animal age/weight?

Lines 101-106: Are you able to include the brand of drugs used?

Line 113: Keep the way brand details are included consistent throughout manuscript.

Section 2.4 (line 128): note here that a single, blinded observer conducted behavioural observations.

Section 2.5 (line 141): More detail required. Can you be more specific with regression models utilised for each analysis? How were between treatment/time differences determined? Was a statistical software package utilised for the analyses? If so, include details.

DISCUSSION

Line 337: Refer to comment above on infrared thermography.

Author Response

Well designed study and well reported. Minor amendments suggested here:

ABSTRACT

Should contain timepoints for measures taken.

Given the limited word count in the abstract, we presented overall effects and not the individual time points.

INTRO

Line 62: Change ‘pain strategy’ to ‘pain management strategy’.

Changed.

Line 65: Infrared thermography isn’t a pain indicator but a method for measuring temperature. Suggest replacing with the outcome that infrared thermography was used to measure.

Changed.

METHODS

Lines 96-97: How was sample size determined and why are animal numbers unequal between treatment groups? Were treatment groups balanced for animal age/weight?

Power calculation added line 100. Reworded to explain the difference in group sizes.

Lines 101-106: Are you able to include the brand of drugs used?

Added line 106.

Line 113: Keep the way brand details are included consistent throughout manuscript.

Only brand name Banamine Transdermal was used. This is now noted on line 98.

Section 2.4 (line 128): note here that a single, blinded observer conducted behavioural observations.

Line 129 - Changed.

Section 2.5 (line 141): More detail required. Can you be more specific with regression models utilised for each analysis? How were between treatment/time differences determined? Was a statistical software package utilised for the analyses? If so, include details.

The model for each outcome was the same as described. More detail is now added starting on line 155. The statistical software information is added on line 143.

DISCUSSION

Line 337: Refer to comment above on infrared thermography.

Changed.

Reviewer 2 Report

Line 46 – states ‘three multimodal pain management protocols.’ Technically, only two of the treatments were multimodal. So, either remove word ‘multimodal’ or correct the statement.  

Line 107-108 – as per the manufacturer’s label of this transdermal product – ‘Do not treat cattle if the hide is wet or may get wet in the six hours after dosing because effectiveness has not been evaluated under wet hide conditions.’ Was this made sure during the study? If yes, that should be stated in this description.

Line 132-133 states the observation of behavior 24 hours after procedure as ‘observed for 5 minutes at a time, every 25 minutes, over a span of two hours. However, line 145-146 and 164 states different info for 24 hour timepoint – ‘one for 4 hours 24 hours after the procedure (5 intervals of 5 minutes each, 25 minutes total.’ This needs to be corrected in line 132-133.

Line 157 – states ‘There was no significant treatment by observation period ..’. Do you mean to state – ‘There was no significant effect of treatment by observation period ….’?

Figure 1 – it is not clear what those listed superscripts represent – a, b, c, and d. Please describe it in text and/or the figure legend.

Figure 2 - it is not clear what those listed superscripts represent – a, b, c, and d. Please describe it in text and/or the figure legend.

For description under 3.2 – can a table or graph be added to share data of salivary cortisol levels for each treatment at all 4 time-points.

Author Response

Line 46 – states ‘three multimodal pain management protocols.’ Technically, only two of the treatments were multimodal. So, either remove word ‘multimodal’ or correct the statement.  

Changed

Line 107-108 – as per the manufacturer’s label of this transdermal product – ‘Do not treat cattle if the hide is wet or may get wet in the six hours after dosing because effectiveness has not been evaluated under wet hide conditions.’ Was this made sure during the study? If yes, that should be stated in this description.

We added this detail to the methods line 109.

Line 132-133 states the observation of behavior 24 hours after procedure as ‘observed for 5 minutes at a time, every 25 minutes, over a span of two hours. However, line 145-146 and 164 states different info for 24 hour timepoint – ‘one for 4 hours 24 hours after the procedure (5 intervals of 5 minutes each, 25 minutes total.’ This needs to be corrected in line 132-133.

Corrected throughout the text. Since the 24-hour time point was actually 25 minutes of observation, this pointed out that the 60 minute (period 3 information) was incorrect and was actually 40 minutes of observation. This resulted in further edits to the results and the figure legends as well as the discussion. Since the rates of behaviors in period 3 decreased, because of the larger denominator (40 min vs 25 min), there are a few significant changes to the results where period 3 is now significantly different from other periods (see new figure 1). There is now also an effect of period on the amount of grooming and postural changes.  There is still no treatment effect or treatment by time interaction for any of the outcomes.

Line 157 – states ‘There was no significant treatment by observation period ..’. Do you mean to state – ‘There was no significant effect of treatment by observation period ….’?

Line 166- Reworded. We meant to state that there were no significant interactions between treatment and observation period for any of the models.

Figure 1 – it is not clear what those listed superscripts represent – a, b, c, and d. Please describe it in text and/or the figure legend.

We added to the figure legend that it represents a significant difference at or below P < .05. Exact values can be found in the text.

Figure 2 - it is not clear what those listed superscripts represent – a, b, c, and d. Please describe it in text and/or the figure legend.

We added to the figure legend that it represents a significant difference at or below P < .05. Exact value can be found in the text.

For description under 3.2 – can a table or graph be added to share data of salivary cortisol levels for each treatment at all 4 time-points.

We elected to include the overall numerical results in the text (line 236) and as a table with the treatment by time values (see table 3).